# Lifelong-Learning Embeddings: Incremental and Continual Representation Learning for Dynamic E-Commerce Trends

## Abstract

E-commerce is a highly dynamic domain where products and consumer behaviors evolve rapidly. Embedding-based representations are central to deep learning–based personalization systems, yet conventional embeddings are static and therefore, they cannot easily incorporate new tokens (e.g., new products) without retraining, which is costly and often infeasible due to privacy or data retention constraints. To address this, we propose Lifelong-Learning Embeddings, a framework that (1) incrementally extends embeddings to integrate new tokens, (2) adapts embedding dimensionality to balance expressiveness and efficiency, and (3) employs continual learning to mitigate catastrophic forgetting. Experiments on a real-world dataset and two benchmark datasets show that our approach consistently outperforms static embeddings in accuracy while incurring only modest training-time overhead, demonstrating its effectiveness and adaptability in dynamic e-commerce environments.

## 1 Introduction

Learning effective representations is a cornerstone of deep learning across domains such as computer vision (CV) (He et al., 2016; Chen et al., 2020) and natural language processing (NLP) (Vaswani et al., 2017; Devlin et al., 2019; Brown et al., 2020), where embeddings play a central role in mapping discrete inputs into continuous spaces that capture semantic and structural relationships. Embeddings enable models to generalize beyond raw symbols, underpinning advances in word representations (Mikolov et al., 2013; Vaswani et al., 2017; Devlin et al., 2019; Brown et al., 2020), image features (He et al., 2016; Chen et al., 2020), as well as node embeddings (Abu-El-Haija et al., 2018; Menand & Seshadhri, 2024).

In e-commerce, embeddings serve a similarly fundamental function. Customer interactions, such as product views, clicks, or purchases, are tokenized into discrete identifiers (e.g., product IDs, URLs) that must be represented in dense vector spaces to power downstream tasks such as recommendation, purchase prediction, or personalization (Alves Gomes et al., 2022; Sheil et al., 2018b; Tercan et al., 2021; Srilakshmi et al., 2022; Vasile et al., 2016; Zeng et al., 2020; Huang et al., 2020; Chen et al., 2022; Zhou et al., 2018). Building on these dense representations, modern deep learning models leverage embeddings to capture evolving customer interests and item relationships, thereby driving state-of-the-art performance in personalization tasks (Zhou et al., 2019; Sun et al., 2019; Chen et al., 2022; Wang et al., 2025; Yang et al., 2024).

However, unlike in CV or NLP, e-commerce environments are inherently dynamic (Ferrera & Kessedjian, 2019). New products are continuously introduced, seasonal campaigns and emerging trends give rise to novel interaction patterns, and entire product segments may rapidly appear or disappear. This constant growth of the item vocabulary poses a major challenge for static embeddings, which typically require complete retraining to accommodate new tokens. Such retraining is computationally costly and memory-intensive, while also prone to catastrophic forgetting, where previously learned knowledge is overwritten by new information because retaining past data is often infeasible due to legal and economic constraints (European-Parliament, 2016; Tallon, 2010; Kiker, 2014; Demirer et al., 2024). At the same time, modern embedding tables in large-scale recommenders can reach terabyte scale, consuming vast resources and slowing training. Existing methods

often allocate uniform embedding sizes regardless of feature frequency, leading to inefficiencies and scalability limits (Luo et al., 2024;?).

This places three critical demands on any embedding mechanism used to support recommendation or prediction tasks in e-commerce:

- **Extendability**: seamlessly integrate new tokens without retraining from scratch.
- **Knowledge Retention**: preserve previously learned representations to mitigate catastrophic forgetting.
- **Compactness**: adapt the dimensionality of embeddings over time to balance expressiveness with computational efficiency.

To address these challenges, we propose Lifelong-Learning Embeddings (LLE), a dynamic embedding framework that integrates incremental vocabulary expansion, adaptive embedding dimensionality, and continual learning (CL) strategies. LLE is designed to incrementally accommodate new tokens while retaining past knowledge and maintaining a resource-efficient representation space. We evaluate LLE on large-scale industrial and benchmark e-commerce datasets, demonstrating consistent improvements over static baselines and providing detailed ablation studies to isolate the contribution of each component.

## 2 RELATED WORK AND ITS LIMITATIONS FOR E-COMMERCE

Foundational methods such as word2vec (Mikolov et al., 2013) and Transformers (Vaswani et al., 2017) have established dense semantic representations as the backbone of NLP. Out-of-vocabulary tokens are usually addressed with placeholder symbols (Sutskever et al., 2014) or subword techniques (Sennrich et al., 2015; Radford et al., 2019; Liu et al., 2019). However, these approaches are ill-suited to e-commerce, where product identifiers are arbitrary, non-compositional, and arrive at high frequency (cf. Figure 2). Unlike natural language, where vocabulary shifts gradually, e-commerce platforms encounter thousands of new items on a continuous basis, rendering static embedding strategies ineffective.

Research into dynamic embeddings has followed two main directions. The first optimizes *embedding dimensions* to reduce memory overhead. For instance, Luo et al. (2024) propose an approach that prunes or extends embedding chunks during training based on access frequency and gradient norms, achieving large memory savings in industrial recommender models. Dong et al. (2024) propose a Bayesian dimension selection framework for embeddings, using adaptive stochastic gradient and sparse regularization to prune irrelevant dimensions. Qu et al. (2024) tackles streaming recommendation by using reinforcement learning to allocate user and item embedding sizes under a strict memory budget, dynamically growing or shrinking embeddings as frequencies change. These works address scalability but not the integration of entirely new tokens.

The second direction studies *dynamic or inductive embeddings*, where representations adapt to evolving interactions. Zhang & et al. (2021) approach models user–item trajectories with graph neural networks, supporting continuous updates and inductive handling of new nodes. Zheng et al. (2021) treats cold-start as a meta-learning task, using a matching network to rapidly form embeddings for new items given a few user interactions. Rossi et al. (2020) provide a general framework for dynamic graphs, where memory modules and message passing allow embeddings to update with each event and to inductively represent new nodes. These methods support new entities through on-the-fly computation or memory updates, but do not maintain persistent embedding tables that expand as in real-world e-commerce systems. Weimann & Conrad (2025) decouple embeddings from the model, using a dynamic cache updated in real time through self-supervised objectives, enabling new items to be added without retraining the whole model. However, the cache is periodically refreshed and filled with newly trained embedding representations, so previously learned representations are not preserved across iterations.

In e-commerce personalization systems, embeddings remain the backbone of click prediction (Zhou et al., 2019; Zhang et al., 2024), ranking (Cheng et al., 2016; Wang et al., 2017; Sun et al., 2019), and behavior modeling (Sheil et al., 2018b; Srilakshmi et al., 2022; Zhang et al., 2023). Yet, most models are retrained on static datasets or periodically refreshed, lacking mechanisms for incremental and continual representation learning. Existing dynamic embedding methods focus either on optimizing

dimension sizes or on inductively generating embeddings, but none provide a principled approach to incrementally expanding embedding representations with new tokens while also preserving past knowledge. This gap is particularly acute in e-commerce, where new products arrive continuously and embeddings must both grow dynamically and adapt continually without retraining from scratch. The LLE framework introduced in this work addresses this need.

## 3 LIFELONG-LEARNING EMBEDDINGS APPROACH

LLE enables dynamic adaptation to incoming data while preserving previously acquired knowledge. The framework consists of three core components as illustrated in figure 1: (1) incremental embedding update, which is able to represent changing token distributions over time, (2) adaptive embedding dimensionality to maintain an efficient and general embedding representation, and (3) training embedding with CL strategies to stabilize the embeddings and prevent drift when training on new distributions. Whereby adding new tokens (1) or extending dimensions (2) does not affect the outputs of models that rely on LLE.

LLE is designed as a drop-in replacement for existing embedding layers, requiring only minimal adjustment of downstream layers if the embedding dimension changes. Each component is self-contained and can be deployed independently. Furthermore, the framework is method-agnostic: initialization strategies, resizing criteria, or CL objectives can be integrated without architectural changes, enabling rapid adoption of future advances. In the following subsections, we describe each component and its integration.

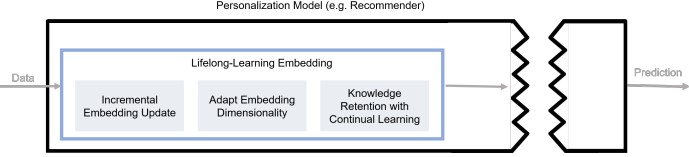

Figure 1: LLE framework: data feed an embedding module with three components: (1) incremental embedding update, (2) adaptive dimensionality, and (3) CL-based knowledge retention. LLE can replace any embedding layer in a prediction model, e.g. recommender.

### 3.1 INCREMENTAL EMBEDDING UPDATE

An embedding layer maps a token from a discrete vocabulary $\mathbb{V}$ to a $d$-dimensional latent space via a matrix $\boldsymbol{E} \in \mathbb{R}^{|\mathbb{V}| \times d}, d \in \mathbb{N}$, where each row encodes one token. The input is typically one-hot, so the layer functions as a lookup table of trainable parameters.

To handle evolving vocabularies, we introduce a dynamic update mechanism. Let $\mathbb{V}^{(t)}$ be the vocabulary at time $t$ with embedding matrix $\boldsymbol{E}^{(t)} \in \mathbb{R}^{|\mathbb{V}^{(t)}| \times d}$, and $\mathbb{V}^{(t+1)}$ the updated vocabulary. The new embedding matrix $\boldsymbol{E}^{(t+1)}$ is defined as:

$$\boldsymbol{E}^{(t+1)}[i] = \begin{cases} \boldsymbol{E}^{(t)}[\pi(i)] & v_i \in \mathbb{V}^{(t)} \\ g(\boldsymbol{E}^{(t)}, v_i) & v_i \notin \mathbb{V}^{(t)}, \end{cases}$$

where $\pi(i)$ maps token indices and $g$ is a function that initializes embeddings for unseen new tokens based on the weights from $\boldsymbol{E}^{(t)}$. For tokens that remain in both vocabularies, this definition guarantees that their embeddings are identical in $\boldsymbol{E}^{(t)}$ and $\boldsymbol{E}^{(t+1)}$, ensuring consistency across updates. The incremental embedding update supports both the growth and shrinkage of $\mathbb{V}$, enabling efficient and adaptive representations over time.

### 3.2 ADAPTIVE EMBEDDING DIMENSIONALITY

Beyond handling evolving vocabularies, embeddings must also adapt their latent dimensionality when the representational capacity of the current space is misaligned with the changing data. Given an embedding matrix $\boldsymbol{E}^{(t)} \in \mathbb{R}^{|\mathbb{V}^{(t)}| \times d}$ at time $t$, we allow both extension and reduction of the latent dimension of the embedding representation.

For an extension to $d' > d$, we allocate a new embedding matrix $\boldsymbol{E}^{(t+1)} \in \mathbb{R}^{|\mathbb{V}| \times d'}$. To preserve prior knowledge, the first $d$ dimensions of each row are copied from $\boldsymbol{E}^{(t)}$, while the newly added $(d' - d)$ dimensions are initialized with zeros:

$$\boldsymbol{E}^{(t+1)}[i][: d] = \boldsymbol{E}^{(t)}[i], \boldsymbol{E}^{(t+1)}[i][d : d'] = 0, \forall v_i \in \mathbb{V}.$$

This zero-initialization to handle new embedding dimensions ensures that the embedding outputs remain unchanged immediately after extension, so the model's behavior is consistent while the new dimensions can be trained gradually. For a reduction to $d' < d$, a new embedding matrix $\boldsymbol{E}^{(t+1)} \in \mathbb{R}^{|\mathbb{V}| \times d'}$ is generated. Since direct truncation would discard information, CL is applied to transfer knowledge from the old higher-dimensional embedding $\boldsymbol{E}^{(t)}$ into the reduced representation.

### 3.3 Embedding Training with Continual Learning

CL provides a stability mechanism that restricts updates for shared tokens ($\mathbb{V}^{(t)} \cap \mathbb{V}^{(t+1)}$) to stay close to their historical representations, effectively controlling semantic drift. This allows the embedding space to adapt to new dimensionalities and new tokens without forgetting past knowledge. Thereby, applying CL to the proposed LLE approach is non-trivial. To make CL work in this setting, we need to solve three issues, since vocabularies may change and embedding dimensions may differ. Therefore, the following three design choices are needed:

1. Restrict knowledge perseveration to tokens present in both vocabularies.
2. Ensure that their indices are correctly matched.
3. Provide a mechanism to compare embeddings of possibly different dimension sizes.

First, we define the overlap of old and new vocabularies as

$$\widetilde{\mathbb{V}} = \mathbb{V}^{(t)} \cap \mathbb{V}^{(t+1)}.$$

Only tokens in $\widetilde{\mathbb{V}}$ are aligned, since new tokens in $\mathbb{V}^{(t+1)} \setminus \mathbb{V}^{(t)}$ have no prior representation.

Second, let $\pi^{(t)}$ and $\pi^{(t+1)}$ denote the index mappings into $\boldsymbol{E}^{(t)} \in \mathbb{R}^{|\mathbb{V}^{(t)}| \times d}$ and $\boldsymbol{E}^{(t+1)} \in \mathbb{R}^{|\mathbb{V}^{(t+1)}| \times d'}$, respectively. For each token $x \in \widetilde{\mathbb{V}}$, its old and new embeddings are $\boldsymbol{E}^{(t)}[\pi^{(t)}(x), :]$ and $\boldsymbol{E}^{(t+1)}[\pi^{(t+1)}(x), :]$.

Third, to handle different embedding sizes ($d' \neq d$), we map both vectors into a common comparison space of dimension $m_x$:

$$\psi_x^{(t)} = P_x^{(t)} \boldsymbol{E}^{(t)}[\pi^{(t)}(x), :], \qquad \psi_x^{(t+1)} = P_x^{(t+1)} \boldsymbol{E}^{(t+1)}[\pi^{(t+1)}(x), :],$$

where $P_x^{(t)} \in \mathbb{R}^{m_x \times d}$ and $P_x^{(t+1)} \in \mathbb{R}^{m_x \times d'}$ are comparison maps (e.g., identity or truncation if $d' = d$, padding for $d' > d$, or projection for $d' < d$).

Finally, the CL alignment loss is then defined as

$$\mathcal{L}_{\text{CL}} = \sum_{x \in \widetilde{\mathbb{V}}} \ell_{\text{align}}\big(\psi_x^{(t+1)}, \psi_x^{(t)}\big),$$

where $\ell_{\text{align}}$ is a general alignment function (e.g., $\ell_2$-norm or cosine similarity).

Our approach is agnostic to the specific CL technique, and any method providing a regularization signal or teacher output from prior embeddings can be used. The total training loss is:

$$\mathcal{L}_{\text{total}} = \mathcal{L}_{\text{task}} + \lambda_{\text{CL}} \mathcal{L}_{\text{CL}},$$

with $\mathcal{L}_{\text{task}}$ the supervised task loss (e.g., cross-entropy) and $\lambda_{\text{CL}}$ a weighting factor. This enables embeddings to retain semantic structure across time while adapting to evolving token distributions. Minimizing $\mathcal{L}_{\text{CL}}$ bounds the drift of overlapping tokens, since for all $x \in \widetilde{\mathbb{V}}$ the deviation $|\psi_x^{(t+1)} - \psi_x^{(t)}|$ is directly penalized. Thus, shared tokens remain anchored to their prior representations while the embedding space adapts to new dimensions and data.

In case of dimension upsizing ($d' > d$), the CL controls the gradients on the old dimensions with regularization so that semantics stay stable, while the new dimensions can gradually absorb additional information from the new distribution. When downsizing ($d' < d$), previous information is

lost and CL will align the new embedding $E^{(t+1)}$ to the old space and reduce semantic loss. Even when the dimensionality stays the same ($d' = d$), CL provides stability by preventing overwriting of established semantics as token distributions shift. Finally, when increasing the token space, CL provides stability by anchoring shared tokens to their previous representations, ensuring minimal drift while the embedding space adapts.

## 4 DATASETS AND PREPROCESSING

We evaluate LLE on one proprietary industrial dataset and two commonly used public benchmarks. The industrial dataset spans 17 weeks and contains 53.8M events across 6.2M sessions. The Yoo-Choose dataset[1], introduced for the RecSys Challenge 2015,[2] comprises 33M events from 9.2M sessions collected over 21 weeks, and has since become a widely used benchmark in the purchase prediction literature (Park et al., 2015; Romov & Sokolov, 2015; Lin et al., 2019; Esmeli et al., 2021; Alves Gomes et al., 2022). The RetailRocket dataset[3] includes 2.75M events from 1.4M users spanning 18 weeks and is frequently adopted for user modeling tasks such as recommendation and churn prediction (Berger & Kompan, 2019; Fridrich & Dostál, 2022; Zeng et al., 2020; Sheil et al., 2018a).

Figure 2 illustrates the token dynamics over time. The datasets differ markedly in interaction volume, the rate of new token introduction, and the prevalence of unknown tokens, which allows for a comprehensive evaluation of LLE under diverse e-commerce patterns and evolving user behavior.

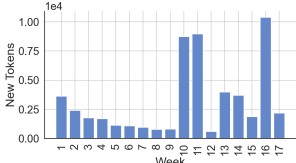
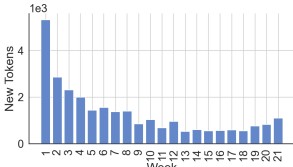
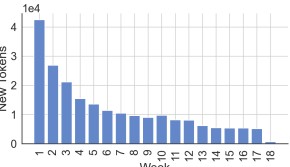

| (a) Number of new tokens per week for Industrial. | (b) Number of new tokens per week for YooChoose. | (c) Number of new tokens per week for RetailRocket. |

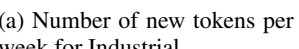
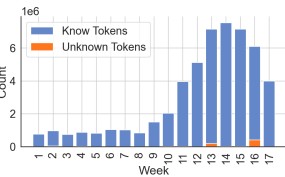
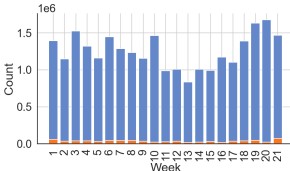
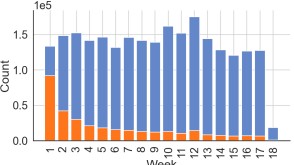

(d) Number of unknown and known tokens per week based on the events for Industrial.

(e) Number of unknown and known tokens per week based on the events for YooChoose.

(f) Number of unknown and known tokens per week based on the events for RetailRocket.

Figure 2: Token statistics over time across the three datasets.

We standardized all datasets into {session ID, interaction object, timestamp}. Sessions in Retail-Rocket were reconstructed by grouping consecutive user interactions with a gap of at most 30 minutes. Sessions shorter than three interactions or longer than the 99.5th percentile were removed. For the industrial dataset, interaction objects were extracted from URLs with query parameters removed to compactify the vocabulary. Customer IDs were used only to derive binary purchase labels (purchase vs. non-purchase) and then discarded.

As reported by Lo et al. (2016), consumer behavior exhibits weekly rhythms such as weekend peaks and payday effects, making this timeframe both behaviorally representative and practically relevant. Therefore, we sliced all datasets into *weekly increments* to simulate incremental learning. Weekly slicing provides a balance between granularity and stability: each batch contains sufficient inter-

---

[1]Available at https://www.kaggle.com/datasets/chadgostopp/recsys-challenge-2015

[2]https://recsys.acm.org/recsys15/challenge/

[3]Available at https://www.kaggle.com/datasets/RetailRocket/ecommerce-dataset

action volume to train embeddings and task-models while avoiding the computational overhead of daily retraining.

# 5 EXPERIMENTAL SETUP

The objective of our experiments is to evaluate whether the proposed LLE method can effectively handle the dynamics of e-commerce characterized by continuously emerging tokens and shifting distributions while retaining past knowledge.

All experiments follow a weekly incremental learning protocol as highlighted in algorithm 1. At each step $t$, the embedding is updated with new data and then evaluated on the subsequent week $t + 1$ in a downstream prediction task. For the experiments, we follow a decoupled design: embeddings are first trained in a self-supervised fashion using context prediction with a cross-entropy loss (akin to Skip-Gram (Mikolov et al., 2013)), and subsequently fed into a Transformer classifier (two layers, adaptive attention heads) for session-level purchase prediction. This setup isolates the contribution of the embedding space itself, avoiding confounding effects of end-to-end fine-tuning. The Transformer model is retrained from scratch at each step to ensure that evaluation reflects only the quality of the updated embeddings, preventing any residual knowledge from earlier models from influencing performance.

---

**Algorithm 1** Dynamic Embedding and Task Model Training

---

1: **for** each new data iteration $D^{(t)}$ **do**
2:     Incremental embedding update
3:     Adapt embedding dimensionality
4:     Train embedding with CL
5:     Train task model with updated embedding
6:     Evaluate task performance on $D^{(t+1)}$
7: **end for**

---

While the embedding and classifier are trained separately, the embedding dimensionality itself remains a design choice that strongly impacts generalization and efficiency. Previous work has explored different strategies for adapting embedding size, ranging from rule-based schedules to validation-driven criteria. In our experiments, we adopt a simple heuristic based on the gap between training and validation losses. Let $\mathcal{L}_{\text{train}}^{(e)}$ and $\mathcal{L}_{\text{val}}^{(e)}$ denote the training and validation losses at epoch $e$. We diagnose overfitting when

$$\frac{\mathcal{L}_{\text{val}}^{(e)}}{\mathcal{L}_{\text{train}}^{(e)}} > \gamma \quad \text{and} \quad \mathcal{L}_{\text{val}}^{(e)} > \mathcal{L}_{\text{val}}^{(e-1)},$$

with divergence threshold $\gamma > 1$. If this condition holds for multiple epochs, the embedding dimensionality $d$ is iteratively reduced by step size $\Delta d \in \mathbb{N}$, $0 < \Delta d < d$ until validation loss improves:

$$d' = d - \Delta d.$$

Conversely, if validation loss continues to decrease without divergence, we increase dimensionality:

$$d' = d + \Delta d,$$

until $\mathcal{L}_{\text{val}}^{(e)}$ no longer improves.

For evaluation, we report AUC scores for each week, training time, and analyze the performance–efficiency trade-off. To measure the impact of the framework, we compare LLE against two reference setups:

- **Baseline 1**: Embedding trained from scratch at each week using only that week's data.
- **Baseline 2 (Upper bound)**: Embedding retrained from scratch at each week but using data from all weeks up to the current point (not feasible in practice but provides an oracle-like reference).

To disentangle the effect of each component in LLE, we conduct a systematic ablation study that varies the following factors:

- **New Token Initialization**: none (retraining), random, unknown-token, or average-based initialization.

- **Dimensionality Adaptation**: with or without adaptive dimensionality adjustment.

- **Continual Learning**: no CL, single regularization-based methods (EWC (Kirkpatrick et al., 2017), MAS (Aljundi et al., 2018), LwF (Li & Hoiem, 2016)), and their combinations.

We focus on regularization-based CL since memory-based replay approaches, while effective (Shin et al., 2017), are limited by scalability, privacy (e.g., GDPR (European-Parliament, 2016)), and storage costs in practice.

All experiments are implemented in PyTorch using AdamW (learning rate $1 \times 10^{-3}$) with a batch size of 128. Each variant is executed independently ten times, and we report averaged results for robust comparison. Unless stated otherwise, the standard LLE configuration employs average-based initialization for new tokens and MAS for CL. Experiments are conducted on a server with 96 Xeon Platinum 8168 CPU cores (2.7GHz) and a single Nvidia Tesla V100 GPU. Although we evaluate LLE in a decoupled setting for clarity, the framework is directly compatible with end-to-end architectures and can be integrated as a drop-in replacement for standard embedding layers.

## 6 RESULTS AND DISCUSSION

Figure 3 shows that LLE consistently outperforms the lower-bound baseline and approaches the upper bound across all datasets. These results indicate that LLE can retain knowledge without access to historical data, which is often limited in practice due to storage costs or privacy constraints. In this evaluation, LLE is applied with incremental updates, average-based initialization for new tokens, and MAS for continual learning, a configuration that yielded a stable and adaptable performance across datasets.

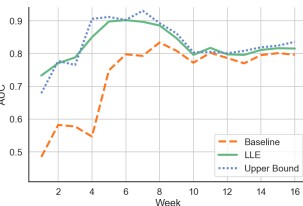 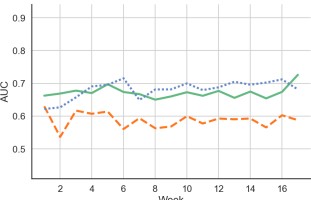

(a) Weekly AUC performance for the industrial dataset.

(b) Weekly AUC performance for the YooChoose dataset.

(c) Weekly AUC performance for the RetailRocket dataset.

Figure 3: AUC results over time for LLE compared to Baseline (lower bound) and Upper Bound across all three datasets.

On the Industrial dataset, LLE clearly outperforms the lower baseline in the first ten weeks and nearly matches the upper bound. After week 10, however, the performance gap narrows as interactions increase from about 1M to over 6M by week 14 (cf. Figure 2), rendering recent data dominant and past information less relevant.

For both YooChoose and RetailRocket, LLE consistently improves over the lower baseline and remains close to the upper bound, despite very different data characteristics. This shows that even under contrasting conditions, LLE effectively leverages past knowledge and adapts to new data.

Figure 4 compares training time. The baseline is the fastest but also the least effective. The upper bound achieves higher accuracy but is the slowest, as it requires full retraining each week with training time scaling linearly in the added data. LLE introduces only modest overhead compared to the baseline while avoiding the escalating costs of the upper bound.

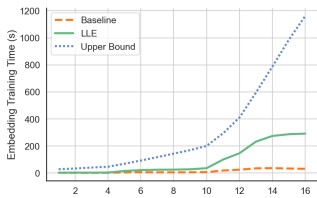 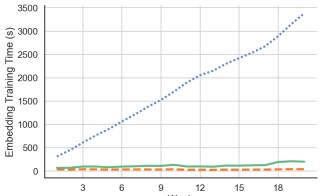 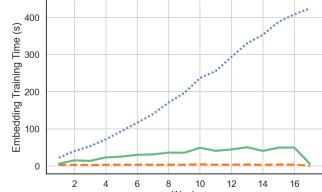

(a) Required embedding train time on the industrial dataset.

(b) Required embedding train time on the YooChoose dataset.

(c) Required embedding train time on the RetailRocket dataset.

Figure 4: Training time per week for LLE compared to Baseline and Upper Bound across all datasets.

The findings demonstrate that LLE achieves a favorable balance between adaptability and efficiency. By incrementally integrating new tokens and stabilizing representations through CL regularization, LLE adapts to evolving data distributions while mitigating forgetting, without the need to retrain on the full history. This design enables LLE to approach the accuracy of the upper bound while maintaining computational costs closer to the lower bound.

## 7 ABLATION STUDY: COMPONENT CONTRIBUTIONS OF LLE

We evaluate the contribution of initialization strategies and CL. The results indicate that the overall design of LLE provides the primary performance gains, while the specific choice of initialization scheme (e.g., unknown-token vs. average) or CL variant (e.g., EWC, MAS, LwF, etc.) acts more as a hyperparameter that needs to be tuned to the needs of a particular use case.

For instance, in YooChoose, the top-performing variants differ only in initialization or CL choice but all achieve mean AUCs around 0.7 over all weeks, which is about +0.07 over the baseline. These findings indicate that the benefit comes from the LLE framework itself rather than from a specific initialization or CL variant.

To further explore this, Figure 5 shows the weekly AUC difference from the baseline across datasets. Thereby, shaded regions indicate variance across runs. We group the approaches into three categories: (1) *Both* incremental initialization and CL (orange), (2) *Only incremental* initialization (blue), and (3) *Only CL* (green).

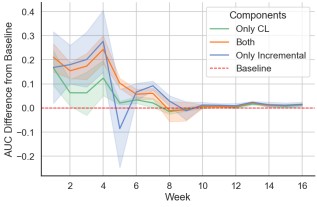 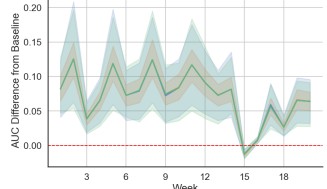 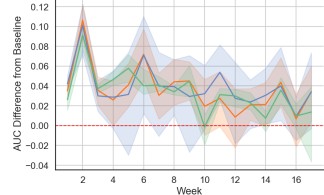

(a) Weekly AUC relative to the baseline for the industrial dataset.

(b) Weekly AUC relative to the baseline for the YooChoose dataset.

(c) Weekly AUC relative to the baseline for the RetailRocket dataset.

Figure 5: Ablation results of using different LLE components on the three datasets.

In the industrial dataset, incremental strategies adapt quickly but show higher variance, CL stabilizes performance but reacts more slowly, and combining both provides the most consistent improvements. The variance across methods decreases after week 10, coinciding with the sharp rise in interaction volumes (cf. Figure 2). This aligns with the earlier performance results (Figure 3a), where the gap between LLE and the baselines narrows once abundant new data dominates. The ablation thus reinforces that when fresh data overwhelms prior knowledge, different strategies converge in performance.

In YooChoose, combining incremental strategies and CL produces smoother performance than either alone, reflecting the benefit of balancing rapid adaptation with knowledge retention. Both mechanisms help, but their combination yields the most stable AUC improvements week by week.

In RetailRocket, the embedding dimensionality is the only one that undergoes shrinkage (cf. Figure 6), which forces the model to discard past information. This explains the high variance of the incremental-only configuration, which at times even falls below the baseline, and the overall limited performance gains. By contrast, CL-only shows the lowest variance and most stable results. This highlights the role of CL in aligning new embeddings with the information of the old space, thereby mitigating the loss induced by dimensionality reduction and stabilizing performance despite the challenging dynamics of this dataset.

As aforementioned, embeddings constitute massive tables, and compact representations are essential for memory efficiency. Adaptive dimensionality plays a complementary role: while larger embeddings may contain all the information needed to solve a task, they are often redundant and costly. Figure 6 illustrates how LLE dynamically regulates capacity across datasets. In RetailRocket, dimensions shrink from ~28 to ~13, indicating that too much capacity was initially allocated, likely due to the influx of new tokens requiring richer representations, which later became redundant and was compressed to prevent overfitting. In the industrial dataset, the opposite pattern emerges. Dimensions start small, but as the event volume grows after week 10, capacity is gradually expanded to avoid under-parameterization. YooChoose remains consistently high (32–35) to accommodate its large and diverse vocabulary. These dynamics show that adaptive resizing does not directly improve AUC but ensures efficiency by allocating capacity where needed and compressing when redundancy dominates.

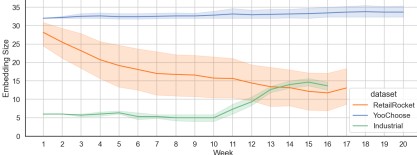

Figure 6: Average embedding size over time.

## 8 CONCLUSION

We introduced Lifelong-Learning Embeddings, a drop-in framework that incrementally integrates new tokens, adapts embedding dimensionality to balance capacity and compactness, and uses CL to retain prior semantics. Across an industrial dataset and two public benchmarks, LLE consistently outperforms training from scratch and approaches the upper bound without requiring access to the full history. The ablation study shows that the specific choice of initialization or CL (EWC/MAS/LwF/combos) has a minor impact on accuracy. As long as some LLE mechanism is applied, gains are robust. Whereby, dimensionality adaptation contributes primarily to efficiency.

## 9 RESEARCH LIMITATIONS AND FUTURE DIRECTIONS

We focused on a weekly, decoupled protocol to isolate representation quality. The behavior under fully end-to-end training or alternative update cadences (e.g., daily or event-driven) remains unexplored in this research. Furthermore, new-token initialization relied on lightweight schemes (random, unknown, average). Richer metadata- or structure-aware approaches could further improve cold-start handling. Our ablation study suggests that the main benefit comes from applying incremental or CL mechanism at all. This highlights LLE's strength in flexibility, but also indicates that gains may plateau without more specialized strategies.

Future work should therefore extend LLE to richer initialization strategies, investigate task-aware or streaming update schedules, and examine integration into end-to-end training regimes.

## Reproducibility Statement

We provide the full implementation of LLE and all experiments in the supplementary material, along with a lightweight Jupyter notebook that highlights the core components of LLE. The notebook can be run on a standard local machine without requiring high-end computational resources.

Publicly available datasets (YooChoose and RetailRocket) can be used to reproduce our results (cf. Section Datasets and Preprocessing). The industrial dataset cannot be released due to confidentiality, but it was included in the experiments to better reflect real-world dynamics.

Upon acceptance, we will release the complete implementation on GitHub to support further use, improvement, and extension by the community.

## GenAI Usage Disclosure

In the preparation of this manuscript, the authors used generative AI tools, including ChatGPT (OpenAI), DeepL Write, and Grammarly, exclusively for language polishing and grammar correction. These tools were not used for generating content, data analysis, experimental design, or drawing conclusions.

The authors affirm that all substantive ideas, algorithms, experimental setups, and analyses presented in this work are original and authored by the listed contributors. We take full responsibility for the integrity, accuracy, and originality of the published content.

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
