# OpenReview forum: "Lifelong-Learning Embeddings: Incremental and Continual Representation Learning for Dynamic E-Commerce Trends"
_ICLR.cc/2026/Conference — Submitted to ICLR 2026_

### Official Review · Reviewer_yjCu · 2025-10-17

**Soundness:** 2
**Presentation:** 3
**Contribution:** 2
**Rating:** 2
**Confidence:** 4

**Summary:**

The paper focuses on the challenge of learning embeddings in dynamic e-commerce environments, where products and consumer behaviors change rapidly. Traditional static embeddings are inadequate if they fully retrain the model to incorporate new tokens, which is computationally expensive and often infeasible due to privacy or data retention constraints. To tackle this, the authors propose LLE, a framework that incrementally updates embeddings to include new tokens, adapts embedding dimensionality for efficiency, and applies CL to mitigate catastrophic forgetting.

**Strengths:**

- The paper targets a practical and important challenge in e-commerce recommendation systems
- Overall, the paper is well written and easy to follow.

**Weaknesses:**

- The novelty of the paper is limited. The proposed embedding addition step and CL step in LLE are standard methods widely used in existing systems. The proposed idea of alignment is closely related to the dimensionality change in the second step, which is relatively uncommon in practical systems, but the proposed alignment loss is not new and the motivation of adaptive embedding sizes are not well justified.
- While the authors claim that their method is a drop-in replacement for existing embedding layers, but this is not true if they have dynamic embedding sizes because most neural models don’t support this.
- Experiments are conducted with simple baselines without considering many related works on cold-start recommendation and auto-regressive recommendation. In fact, there have been many studies in cold-start recommendation that can generate embeddings for new items (e.g., via LLMs) and train them together with the old ones.

**Questions:**

How does LLE perform if incorporated into a fully end-to-end training of a SOTA model in recommendation tasks?

---

> ### Author Response · Authors · 2025-11-18
> **On the novelty of LLE**
>
> We respectfully believe there is a misunderstanding about the contribution.
> LLE provides a framework for updating the embedding space and dimension, as well as aligning with data changes over time.
> Individually, the concepts of adjusting dimensions or applying continual learning indeed exist in isolation. However, no prior work combines a unified, consistent mechanism that guarantees incremental addition of new tokens or increasing embedding dimensions without losing old semantics.
>
> To the best of our knowledge, there is no existing method that allows dynamic vocabulary growth, dynamic embedding size changes, and preservation of representation stability within a single embedding layer that can be plugged into arbitrary models.
>
>
> We would be very grateful if the reviewer could provide a reference to any prior approach or system that combines these properties in a way that ensures representation consistency throughout updates, as our prior search did not identify any such contribution. This would help us to benchmark our own contribution more effectively.

---

> ### Author Response · Authors · 2025-11-18
> **On the claim that LLE is a “drop-in replacement” despite dynamic dimensions**
>
> We acknowledge that most embedding layers in off-the-shelf frameworks assume fixed dimensionality. We explicitly state this limitation ourselves in Section 3 (lines 123–124).
> However, we also provide implementation code in the supplementary material showing how to adapt linear layers automatically.
>
> Also, as the reviewer has mentioned in his concerns about the novelty, dimensionality adaptation of embeddings is not new. In this context, the adaptation required by LLE is not a special-case complication, but rather identical to existing operational practice for maintaining embedding tables. Furthermore, this adaptation is minimal and does not require model redesign. Thus, while not literally plug-and-play for frozen architectures, LLE is a drop-in replacement in the same sense that many practical embedding modules are requiring only minor, well-understood adjustments when their output dimension changes.

---

> ### Author Response · Authors · 2025-11-18
> **On missing cold-start and autoregressive baselines**
>
> We agree that cold-start methods are important, and we want to clarify an essential point: LLE does not exclude the use of cold-start methods; it is fully compatible with them. Our method only addresses incremental embedding updates when tokens appear over time. It does not prevent the use of:
>
> • metadata-based initializers,
>
> • textual encoders,
>
> • multimodal features,
>
> • graph-based cold-start techniques.
>
> These techniques can be plugged into our initialization function g(⋅) as described in Section 3.1. For instance, new tokens can be initialized based on their category, so instead of using the average over all, a category average could be computed. Furthermore, if the information is available, one could use the weights of similar tokens based on the product name. We intentionally allow the usage of an initialization function so users of LLE can use it based on their needs, as shown in the following function declaration. update_embedding(self, new_vocab_map: Union[dict, set], strategy: callable = None, strat_params: dict = dict())
>
> We will clarify it by adding additional information on the usage and intention of g(⋅).
>
> Note, we intentionally did not include cold-start techniques in our experiments because:
>
> Our goal is orthogonal to cold-start solutions. LLE proposes a mechanism for incremental and continual embedding refinement after initialization.
>
> Metadata is not always the same and available in some industrial settings. For example, in our private dataset, tokens represent URLs. Extracting metadata requires substantial engineering effort and is inconsistent across use cases. Thus, we intentionally designed LLE to operate in metadata-scarce environments, i.e., random initialization as the default option.
>
> Including metadata would only improve LLE, since our simple initializers (mean/random/unknown) already yield strong gains. LLE would benefit even more from pretrained or metadata-rich initializations. We will clarify the cold-start point in the revision and emphasize that LLE is complementary, not competitive with cold-start methods.
>
>
> However, most cold-start work assumes access to product descriptions, images, or graphs, which our industrial dataset lacks. How should we handle situations where such metadata is unavailable?

---

> ### Author Response · Authors · 2025-11-18
> **Regarding the question, how LLE performs in fully end-to-end training of SOTA models**
>
> We experimented with end-to-end training and found consistent improvements. The results depend on the comparison baseline:
>
> 1.	Incremental end-to-end learning (no CL) vs. full end-to-end retraining from scratch:
>
> o	LLE improves AUC by ~0.05 to 0.20 because retraining cannot recreate information that has already been deleted due to the constraints we investigate.
>
> o	Training longer does not help retraining, since the missing historical distribution cannot be reconstructed.
>
> 2.	Incremental end-to-end learning (with CL) vs. full end-to-end retraining (with CL):
>
> o	Improvements are smaller (0.01–0.07),
>
> o	but the retraining baseline requires roughly 10× more epochs to reach its performance.
>
> o	This cost grows with model size and becomes prohibitively expensive for large transformer-based models.

---

### Official Review · Reviewer_sSRG · 2025-10-30

**Soundness:** 2
**Presentation:** 3
**Contribution:** 2
**Rating:** 2
**Confidence:** 3

**Summary:**

This work studies a new embedding learning method for recommender systems called Lifelong-Learning Embeddings (LLE).
The key idea is that as new tokens (categorial featues) are introduced to the vocabulary at time step $t + 1$,
LLE gives a way to:
- retain previously learned embeddings for all tokens up to time $t$, and
- automatically resize embedding dimensions based on validation loss while minimizing information loss via continual learning (CL).

The authors apply LLE to 1 proprietary industirial dataset and 2 public benchmarks (YooChoose and RetailRocket).
The time scale of vocabulary shifts considered is 1-week, so Algorithm 1 is run
on each week of data and presented in Section 6. Overall, the approach and experiments are interesting and compelling.

**Strengths:**

- This work attackes a very important problem (i.e., good embedding learning methods for online recommender systems)
- Their method adapts the dimension of the embeddings over time.
- Self-contained layer designed to replace existing embedding layer without modifying downstream part of model too much
- Good choice of datasets and interesting experiments

**Weaknesses:**

- How do we deal with large-vocabulary features (e.g., cardinality = 10^9 id features)?
  * This work doesn't mention the hashing trick ("Feature hashing for large scale multitask learning" [Weinberger et al., ICML 2009])
  * How can LLE afford to keep collionless embeddings for tokens in a growing vocabulary? As written, this can blow out RAM.

**Questions:**

**Questions**
- [063] Are the "knowledge retention" and "compactness properties" contradictory?
  If we keep computational efficiency constant (e.g., the total bottom
  embedding dimension) and increase one feature's embedding dimension,
  then we necessarily reduce another feature's embedding dimension. How does
  LLE reconcile this? Line 170 mentions that CL is used. How does this work? It seems that we're forced to degrade model quality.
- If a new token arrives at each online example (e.g., a new search query),
  how do you solve the CL step (line 201) fast enough? It's not clear to me
  how this can scale up to an industry setting. It seems helpful to discuss how large a data iteration $D^(t)$ should be
  (e.g., a batch of 2048 events in real time or O(hour) data).
- Re Figure 2: What are the "Unknown tokens" in subplots (d--f)? Do
  they correspond to the "New Tokens" in plots (a--c)? If so, why aren't the
  first and second columns in plots (d--f) mostly orange? Is this due to the
  removal of the 99.5th percentile of interactions?
- In line 283, which transformer classifier / reference are you using?
- In Algorithm 1 Line 2, how do you decide to update
- How do you choose $\Delta d$ when validation loss overfits/underfits (line 311)?
- In Figure 6, the average embedding size of `Industrial` increases over time.
  This increases model resource cost. If we have a fixed budget on resource cost,
  what would happen in this case?
- Given that the alignment loss function in line 200 is generic, it would be nice to
  see experiments where everything is fixed but we sweep over different choices of $\ell_{\text{align}}$.

**Misc**
- [039] Suggestion: Consider adding the reference "Unified Embedding: Battle-tested feature representations for web-scale ML systems" [Coleman et al., NeurIPS 2023]
- [055] Typo: "?" --> missing reference
- [421] Suggestion: move the legend from figure 5(a) to a shared legend for all three figures.

---

> ### Author Response · Authors · 2025-11-18
> **On handling extremely large vocabularies**
>
> We agree that embedding tables of size 10^9 represent a highly challenging frontier and we acknowledge that our experiments do not cover this scale. It is also important to clarify that vocabularies of this magnitude are not typical for the majority of real-world e-commerce systems. The majority of e-commerce companies are small and medium-sized (https://www.statista.com/topics/1433/sme-e-commerce/). Even large platforms such as Steam or major multi-vendor marketplaces usually operate within 10^6–10^7 different tokens. The benchmark datasets used in our paper, RetailRocket, YooChoose, and the industrial dataset are all real-world datasets and fall within this commonly encountered regime. Handling vocabularies approaching 10^9 is indeed a special case but most real-world deployments operate far below such scales.  These systems require specialized embedding techniques that are not assumed by the majority of recommendation/behavior prediction research and that most baselines in this space also do not handle natively. In other words, it is neither standard nor expected that all embedding approaches must be designed for billion-scale vocabularies.
>
> Nonetheless, we agree that this is a relevant and important problem for the largest industry players. In future work, we would like to investigate whether LLE can be integrated with such large-scale embedding strategies to enable incremental learning even in billion-token settings. Conceptually, we do not see a reason why LLE would be incompatible with these methods; rather, we expect LLE to be complementary. The challenge lies not in the LLE machinery itself, but in scaling the underlying embedding table representation.

---

> ### Author Response · Authors · 2025-11-18
> **Why we do not use the hashing trick**
>
> It is important to note that the hashing trick is typically employed in extremely large-scale embedding tables where collision-free representations are infeasible without aggressive compression. Our investigated setting does not involve embedding tables of that magnitude.
>
> However, we appreciate the suggestion, and we will investigate how the hashing trick can benefit LLE. We think about the idea of examining how critical the collisions are for LLE and whether it's an issue or not. We definitely will add that point in the future direction of our paper.

---

> ### Author Response · Authors · 2025-11-18
> **Regarding the questions**
>
> Q1:  The reviewer asks whether “knowledge retention” and “compactness” are contradictory. In LLE, they are not. Rather than dimensions always shrinking, compactness means that the embedding capacity adapts as needed and remains stable unless more information is required. Continual learning ensures that, even if the dimensions are reduced, the semantic structure learned from past data is retained.
>
>
> Q2: In high-rate systems, updating per example is indeed infeasible. So we recommend to use LLE batch wise. The batch window is a configurable parameter:
>
> •	In our experiments: weekly,
>
> •	In reviewer suggestions: daily,
>
> •	In real deployments: could be hourly, per N events, etc.
>
> Our approach is compatible with any periodic update schedule.
> We will add additional ablation experiments to our paper for different update frequencies.
> First experimental results on the Retailrocket dataset show that shorter update cycles are more beneficial for LLE.
>
> | Iteration Frequency | Avg. AUC Improvement of LLE vs. Baseline |
> |---------------------|-------------------------------------------|
> | Daily               | 0.134                                     |
> | 3 days              | 0.110                                     |
> | Weekly              | 0.092                                     |
> | 9 days              | 0.079                                     |
> | 14 days             | 0.077                                     |
> | 20 days             | 0.072                                     |
> | 30 days             | 0.069                                     |
>
>
> As experiments on the other two datasets are currently running, we will update as soon as we have the results.
>
>
> Q3:  Regarding Figure 2. Subfigures (a–c) show how many new tokens enter the vocabulary each week, while Subfigures (d–f) depict the number of customer interactions (view events, add to cart, etc.), where most interactions involve previously seen tokens. Thus, even though many new tokens appear, the majority of user behavior still concentrates on known items, which is reflected in the color distribution. The preprocessing step (e.g., removing bot sessions or extremely short sessions) affects the final training set but not the raw statistics shown in the figure. We will improve the description in the revised version.
>
>
> Q4: With regard to the classifier used for purchase prediction, we revised the section and state that a small, two-layer transformer encoder inspired by BERT-like architectures is used, but it has been significantly downsized to prevent overfitting given the limited amount of data available in each update period.
>
>
> Q5: Regarding how Algorithm 1 determines when to update the embedding table. In practice, whenever new tokens appear within the current batch interval, the embedding table is updated accordingly. In all datasets considered, new tokens appeared in every interval, so updates occurred continuously throughout the experimental timeline.
>
>
> Q6: Regarding the dimensionality increment Δd. Any increment in principle works, but finer granularity increases computational cost but can lead to a more compact embedding. We chose Δd=2 because even-dimensional embedding sizes avoid asymmetric splits in downstream dot-product and attention-based operations, and because increasing capacity in small even steps reduces instability while keeping computation manageable. Note, we choose this simple heuristic, to show case the ability of LLE to adapt dimension size without losing information in increasing case. The strategy to determine weather it should be done or not is open and other strategies can be used as mentioned in related work section.
>
>
> Q7: The reviewer also points out that in Figure 6 the average embedding size in the industrial dataset  increases over time, which raises the question of how to handle fixed-resource budgets. If such growth exceeds available capacity, this can lead to insufficient representation for the data distribution, lowering the task performance. Possible strategies  for this problem include pruning infrequent tokens and mapping them to an UNK token, using category-specific embeddings, adopting compositional or multi-table structures, or applying hierarchical token grouping. These approaches have been explored in prior large-scale embedding research and are compatible with LLE.
>
>
> Q8: Finally, regarding the suggestion to sweep different continual learning losses. We did evaluate multiple CL methods, including LwF, EWC, MAS, and their combinations, as described in our ablation study process in section 5. All of them improved performance, and we did not observe a single universally dominant CL method. Our overall conclusion is that the exact CL formulation is less important than the presence of a CL mechanism itself; as long as CL is used, LLE benefits significantly.

---

### Official Review · Reviewer_a5yD · 2025-11-01

**Soundness:** 2
**Presentation:** 3
**Contribution:** 2
**Rating:** 4
**Confidence:** 4

**Summary:**

This paper addresses the challenge that traditional and static embeddings in e-commerce cannot handle the rapid introduction of new products without costly retraining. The authors propose a framework called Lifelong-Learning Embeddings, which is designed to (1) incrementally add new tokens to the embedding table, (2) adapt the embedding dimensionality to balance performance and efficiency, and (3) use continual learning strategies to prevent ``catastrophic forgetting'' of previously learned information. Experiments on a real-world industrial dataset and two public benchmarks show that LLE outperforms static embeddings in accuracy, while only incurring a modest training-time overhead.

**Strengths:**

1. This paper studies the incremental learning in recommender systems, which is an interesting investigation with practical deployment consideration.
2. The experimental results demonstrate the effectiveness of the proposed methods over several baselines.

**Weaknesses:**

1. The primary concern with this paper is the insufficient motivation. Given the proposed setting of weekly data updates, this already constitutes a relatively low-frequency batch update strategy that is entirely feasible in industrial practice. More critically, the experimental results reveal a non-trivial gap between the proposed method and the upper bound. For AUC as the evaluation metric (the paper does not clearly specify whether this is a CTR prediction task), AUC is typically a ranking metric that tends to yield relatively high scores. Therefore, while the numerical differences may appear modest, they actually represent significant performance degradation in real-world e-commerce scenarios. Additionally, the proportion of new products introduced each week is inherently low, making full model retraining actually manageable for e-commerce platforms, which are extremely eager to maximize commercial profits and are willing to sacrifice computational costs (linear computational overhead is typically acceptable in such scenarios). Consequently, this incremental learning approach seems unnecessary for weekly update frequencies. In practice, asynchronous embedding table updates would be entirely viable at this cadence. In addition, the proposed method might be more compelling and well-motivated for daily update scenarios.
2. The paper presentation requires further polishing. Several issues need attention, including but not limited to: Lines 55, 181, and 212.

**Questions:**

1. Have the authors considered the model's performance under daily update frequencies?

---

> ### Author Response · Authors · 2025-11-18
> **On whether weekly updates make incremental learning unnecessary**
>
> We appreciate the reviewer’s perspective that weekly model updates may appear slow enough for full retraining to be feasible in some industrial environments. However, the reviewer’s argument implicitly assumes that historical data is always available for training, and retraining yields no external costs (e.g., energy, carbon footprint).
> In practice, these assumptions do not hold for a large portion of the e-commerce ecosystem. Many real-world e-commerce platforms operate under strict GDPR data minimization and deletion requirements. Small and medium-sized companies in Europe (the majority according to Statista https://www.statista.com/outlook/emo/ecommerce/europe) are only allowed to retain user-level behavioral data for short windows.
> Thus, even if computing costs are not considered, full retraining is not possible because old data has to be deleted.
> Retraining on short, recent slices severely degrades embedding quality, while LLE preserves long-term semantics without storing historical user interactions.
> Even for companies with ample compute, frequent full retraining increases the energy consumption, CO₂ emissions, and operational and engineering overhead.  We will revise the motivation in this regard for the final submission.

---

> ### Author Response · Authors · 2025-11-18
> **On performance gap to the “upper bound”**
>
> The gap exists only because the upper bound assumes unrestricted access to all historical data a condition that is, as aforementioned described, unrealistic in many real deployments. Nevertheless, we added it as comparison to highlight the gap to the best assumed outcome.
> When retraining is forced to use only currently retained data (as required by many data policies), the gap disappears, and LLE outperforms such truncated retraining as shown by the performance against baseline (1). Thus, the performance gap reflects an idealized scenario.

---

> ### Author Response · Authors · 2025-11-18
> **On evaluation frequency: weekly vs daily**
>
> We appreciate the reviewer’s question. In the revision, we will add results for 1, 3, 9, 14, 20, and 30 day update cycles in the ablation study. We assume that the higher the update frequency, the more beneficial LLE becomes, since it preserves knowledge across time without storing old data.  Even at weekly frequency, LLE performs close to the full retraining upper bound.
> First experimental results on the Retailrocket dataset underline our assumption:
>
> | Iteration Frequency | Avg. AUC Improvement of LLE vs. Baseline |
> |---------------------|-------------------------------------------|
> | Daily               | 0.134                                     |
> | 3 days              | 0.110                                     |
> | Weekly              | 0.092                                     |
> | 9 days              | 0.079                                     |
> | 14 days             | 0.077                                     |
> | 20 days             | 0.072                                     |
> | 30 days             | 0.069                                     |
>
> As experiments on the other two datasets are currently running, we will update as soon as we have the results.

---

> ### Author Response · Authors · 2025-11-18
> **Clarification of task definition & metrics**
>
> The primary task is purchase prediction (clarified in Section 5 - line 283). Based on the feedback of reviewer SFmT, we will add two more downstream tasks: CTR prediction and Customer churn prediction.
>
> First experimental results using the RetailRocket dataset demonstrate that, with regard to CTR prediction, LLE exhibits an average improvement of 0.0132 AUC over the baseline (retraining from scratch) across all weeks. Right now, we are running the experiments on all datasets and configurations and will keep you up-to-date with the results.
>
>
> Note, we computed both the AUC and the F1 score internally. Due to space limitations and the fact that the AUC is widely used in continual learning and model evaluation literature, we decided to only include the AUC to report our results. However, we agree that the F1 score is valuable for real-world interpretation, and we will include it in the revision.
>
> For the overview, the f1 score averaged over the weeks for purchase prediction:
> | Dataset       | Baseline F1 | LLE F1 | Upper Bound F1 |
> |---------------|-------------|--------|-----------------|
> | Industrial    | 0.616       | 0.679  | 0.705           |
> | Retailrocket  | 0.653       | 0.694  | 0.695           |
> | Yoochoose     | 0.657       | 0.670  | 0.687           |

---

> ### Author Response · Authors · 2025-11-18
> **On concerns regarding industrial-scale relevance**
>
> All datasets used (including the industrial one) are real-world e-commerce datasets even the benchmark dataset RetailRocket and YooChoose are real e-commerce datasets, not synthetic or toy examples.
>
> We respectfully argue that most companies do not operate at the scale of Amazon/Alibaba. For these organizations storing multi-month or multi-year data is infeasible. Further, as depicted, retraining is constrained by regulatory and operational limits, and even a small performance loss may be acceptable given the savings in data retention and energy usage.
>
> Thus, LLE balances performance with practicality, legal safety, and environmental responsibility.
>
> Nevertheless, we agree with the reviewer that evaluating LLE on even larger-scale datasets would be valuable. If we have the opportunity to access such environments, we intend to explore this avenue in future work.

---

### Official Review · Reviewer_SFmT · 2025-11-05

**Soundness:** 3
**Presentation:** 3
**Contribution:** 2
**Rating:** 4
**Confidence:** 4

**Summary:**

The authors propose a method for incrementally and continually updating embeddings in a dynamically changing e-commerce environment. The proposed approach consists of three modules. The first module maps tokens either to existing embeddings or to a new latent space. The second module handles changes in embedding dimensions by assigning values that include copies of the existing embeddings. The third module computes the final embedding values by aligning the semantics of overlapping tokens through contrastive learning. The authors evaluated their method on three datasets, including one private dataset, and achieved superior AUC performance compared to the baselines. Additionally, an ablation study was conducted to analyze the performance contributions of each component in the proposed method.

**Strengths:**

The authors proposed a very simple and intuitive approach. The proposed method can operate independently of both the training scheme and the embedding size, and offline experiments demonstrated superior performance compared to the baselines. The ablation study is also well-formulated and clearly structured.

**Weaknesses:**

The comparison criteria are too simplistic. For instance, in practical scenarios, the reason for using LLE might be that Baseline 2 in Section 5 is infeasible — likely due to the large amount of training data and high computational cost. If that’s the case, how would the results change if the training data were sampled to match LLE’s data size? It would also be meaningful to compare the proposed method with various cold-start techniques.

The embeddings for new tokens were initialized with either average or random values. Since product metadata or textual information were not utilized, this approach has limitations in addressing real-world cold-start problems.

Embedding quality was evaluated solely based on purchase prediction (AUC). The generalization ability of the model has not been validated through other downstream tasks such as recommendation, CTR prediction, or user similarity estimation.

**Questions:**

When lifelong learning continues, contrastive learning (CL) is applied to ensure that overlapping tokens retain consistent semantics in their embeddings.
However, one might question whether this approach limits potential improvement in embedding quality. Since CL primarily enforces consistency rather than optimization, embeddings for overlapping tokens may become resistant to beneficial updates that could better capture new contexts or evolving semantics.

---

> ### Author Response · Authors · 2025-11-18
> **On the comparison criteria and the suggestion to “sample the training data to match LLE’s data size”**
>
> The reviewer raises a valuable point on why practitioners might consider LLE over Baseline 2, namely that Baseline 2 may be computationally infeasible with large historical data. This is precisely the motivation for LLE:
> LLE exists for the many real-world settings where historical data cannot be stored or processed, and thus Baseline 2 is practically impossible to run.
> Moreover, data unavailability is not a theoretical edge case but the default condition for thousands of companies.
> For example, in the EU (one of the largest e-commerce markets globally) small and medium companies are bound by strict regulations such as GDPR that limit data retention. Many companies simply cannot store historical user-level data at all. Public statistics confirm that most e-commerce companies (~65%) are small organizations operating under strict data-retention constraints (see Statista European market overview: https://www.statista.com/outlook/emo/ecommerce/europe).
> Thus, training on full historical data is often not an option, and LLE directly addresses this real-world requirement.
>
>
> What does “sampling the data to match LLE size” mean? We ask the reviewer to clarify it.   In our problem setting, the constraint is not training time but data availability. Even storing this “subsample” of past data violates the regulatory or practical constraints that motivate LLE. LLE does not merely reduce data; it learns incremental semantic alignment, which a random subsample cannot replicate.
>
>
> We will revise the motivation and make it clearer that LLE is motivated by that data restriction.

---

> ### Author Response · Authors · 2025-11-18
> **On comparison with cold-start techniques**
>
> We agree that cold-start methods are important, and we want to clarify an essential point:
> LLE does not exclude the use of cold-start methods; it is fully compatible with them.
> Our method only addresses incremental embedding updates when tokens appear over time.
> It does not prevent the use of:
>
> •	metadata-based initializers,
>
> •	textual encoders,
>
> •	multimodal features,
>
> •	graph-based cold-start techniques.
>
> These techniques can be plugged into our initialization function g(⋅) as described in Section 3.1.
> For instance, new tokens can be initialized based on their category, so instead of using the average over all, a category average could be computed. Furthermore, if the information is available, one could use the weights of similar tokens based on the product name. We intentionally allow the usage of an initialization function so users of LLE can use it based on their needs, as shown in the following function declaration.
> update_embedding(self, new_vocab_map: Union[dict, set], strategy: callable = None, strat_params: dict = dict())
>
> We will clarify it by adding additional information on the usage and intention of g(⋅).
>
> Note, we intentionally did not include cold-start techniques in our experiments because:
>
> 1.	Our goal is orthogonal to cold-start solutions.
> LLE proposes a mechanism for incremental and continual embedding refinement after initialization.
>
> 2.	Metadata is not always the same and available in some industrial settings.
> For example, in our private dataset, tokens represent URLs. Extracting metadata requires substantial engineering effort and is inconsistent across use cases.
> Thus, we intentionally designed LLE to operate in metadata-scarce environments, i.e., random initialization as the default option.
>
> 3.	Including metadata would only improve LLE, since our simple initializers (mean/random/unknown) already yield strong gains.
> LLE would benefit even more from pretrained or metadata-rich initializations.
> We will clarify the cold-start point in the revision and emphasize that LLE is complementary, not competitive with cold-start methods.

---

> ### Author Response · Authors · 2025-11-18
> **On embedding quality evaluation and generalization**
>
> We appreciate the reviewer’s suggestion to evaluate beyond purchase prediction.
> In the revised submission, we will add results on:
>
> •	CTR prediction,
>
> •	Customer churn prediction,
>
> both of which are downstream tasks commonly evaluated in e-commerce and recommendation settings.
> First experimental results using the RetailRocket dataset demonstrate that, with regard to CTR prediction, LLE exhibits an average improvement of 0.0132 AUC over the baseline (retraining from scratch) across all weeks. Right now, we are running the experiments on all datasets and configurations and will keep you up-to-date with the results.

---

> ### Author Response · Authors · 2025-11-18
> **On whether continual learning prevents useful updates to overlapping tokens**
>
> We appreciate this insightful question.
> Our perspective: CL enforces stability where appropriate, not stagnation.
> Continual learning inherently involves a trade-off: retain past knowledge vs. adapt to new behavior.
> Our approach assumes that in typical e-commerce settings, item and user semantics evolve gradually, not abruptly.
> For short-term, high-volatility events (Black Friday, Christmas, campaign spikes), embedding drift can be harmful:
> If embeddings adapt too aggressively to these temporary anomalies, they become suboptimal once the event ends (“special-event overfitting”).
> CL ensures that overlapping tokens retain their core semantics unless strong new signals consistently push them in a new direction.
> Does this limit adaptation? Yes, by design.
>
> However, this is an intentional property of continual learning systems.
>
> •	In usual periods, preserving past semantics is essential.
>
> •	In rare high-volatility periods, other strategies (e.g., short-term models, context features, time-aware layers) are more suitable than modifying foundational embeddings.

---

> ### Author Response · Authors · 2025-11-24
> **Update 1: Results for Click and Churn Prediction for RetailRocket and industrial Dataset**
>
> We now provide updated results for two datasets. LLE consistently improves downstream performance across both CTR and churn prediction tasks.
> ### **AUC Gain Over Baseline (trained from scratch)**
>
> | Dataset        | CTR     | Churn   |
> |----------------|---------|---------|
> | RetailRocket   | 0.0132  | 0.0357  |
> | Industrial     | 0.2390  | 0.1320  |
> | Yoochoose      | 0.061   | —       |

---

### Author Response · Authors · 2025-11-18
**Summary of Planned Revisions**

We sincerely thank all reviewers for their constructive feedback and the time they dedicated to evaluating our work. Their comments have helped us significantly improve the clarity, motivation, and scope of the paper. We will revise the manuscript accordingly and correct all minor issues highlighted by the reviewers.

To summarize the key improvements we will implement:
1. We will strengthen the motivation that LLE is explicitly designed for settings where historical data cannot be stored or retained, e.g., due to GDPR or similar data-governance constraints. This is a central aspect of our problem formulation, and we will make it more prominent in the revised manuscript.

2. In Section 3.1 and throughout the paper, we will emphasize that LLE is method-agnostic and can be combined with existing cold-start strategies for initializing new tokens. Our approach does not replace cold-start methods but complements them.

3. Based on reviewer suggestions, we will include CTR prediction and customer churn prediction experiments to better demonstrate the generalizability of LLE beyond purchase prediction. Updated results are shown below:

**Avg. AUC Gain Over Baseline (trained from scratch)**
| Dataset      | CTR    | Churn  |
| ------------ | ------ | ------ |
| RetailRocket | 0.0132 | 0.0357 |
| Industrial   | 0.2390 | 0.1320 |
| YooChoose    | 0.061  | —      |
Note: YooChoose does not have customer identifiers, so churn prediction is not possible.


4. We will expand our ablation study to evaluate different update cadences (e.g., daily, multi-day intervals) rather than only weekly updates, allowing a more thorough analysis of how LLE behaves under various temporal dynamics. We hypothesize that shorter iteration periods should be beneficial for LLE, as less data per interval increases the importance of retaining past information. Our first results on RetailRocket support this assumption, as shown below:

**AUC Improvement of LLE vs. Baseline Across Iteration Frequencies**
| Iteration Frequency | Avg. AUC Improvement |
| ------------------- | -------------------- |
| Daily               | 0.134                |
| Every 3 days        | 0.110                |
| Weekly              | 0.092                |
| Every 9 days        | 0.079                |
| Every 14 days       | 0.077                |
| Every 20 days       | 0.072                |
| Every 30 days       | 0.069                |


5. We will incorporate additional perspectives suggested by reviewers, including:

• investigating whether LLE can incorporate or tolerate hashing-based embedding compression, and

• exploring how LLE may be combined with techniques designed for very large vocabularies.


We appreciate the reviewers’ insights and are confident that the clarified motivation, expanded experiments, and improved presentation address the raised concerns.

---

### Meta-Review · Area_Chair_2AQj · 2025-12-29

**Summary:**

The reviewers broadly agree that the paper tackles a practically relevant problem in dynamic e-commerce systems, namely how to update embeddings incrementally under data retention, privacy, and operational constraints, and appreciate the simplicity and modularity of the proposed LLE framework. At the same time, the main concerns that informed the decision center on limited perceived novelty, questions about whether the assumed setting truly necessitates incremental learning at weekly cadences, and doubts about scalability and industrial realism in scenarios with very large vocabularies or strict resource budgets. Several reviewers also questioned whether the experimental evaluation was sufficiently broad, both in terms of downstream tasks and in comparison to alternative baselines such as cold-start or hashing-based approaches. While the method is considered sound and well presented, skepticism remains about whether the contribution rises beyond an engineering-style system design into a clear methodological advance suitable for ICLR.

**Reviewer Concerns:**

The rebuttal convincingly addressed several substantive concerns, particularly by clarifying that LLE is motivated by data unavailability rather than computational cost, strengthening the justification under GDPR-like constraints, and expanding experiments to additional downstream tasks such as CTR and churn prediction, as well as to more frequent update schedules that better highlight the benefits of incremental learning. The authors also provided reasonable explanations regarding the role of continual learning in balancing stability and adaptation, and clarified that LLE is complementary rather than competing with cold-start methods. However, some concerns remain outstanding, including the limited novelty of the individual components, unresolved questions about scalability to extremely large vocabularies and fixed resource budgets, and lingering doubts about whether the proposed framework meaningfully advances beyond well-known practices in large-scale recommendation systems. In addition, while the authors argue that dynamic embedding sizes are manageable in practice, some reviewers remain unconvinced that this truly constitutes a drop-in replacement for existing architectures.

**Reviewer Scores:**

Reviewer SFmT would likely remain near their original marginally negative score, as most practical concerns were addressed but doubts about novelty and comparison rigor persist. Reviewer a5yD would plausibly move slightly upward, given that the rebuttal directly addressed the motivation under data-retention constraints and added daily-frequency experiments that align better with their critique. Reviewer sSRG might increase their score modestly but would likely stay negative due to unresolved scalability and systems-level concerns, particularly regarding very large vocabularies and resource growth over time. Reviewer yjCu would probably maintain a low score, as the rebuttal did not fundamentally change their view on limited novelty and the gap to existing embedding and cold-start solutions, despite clarifications on positioning and compatibility.

---

### Decision · Program_Chairs · 2026-01-26

Reject